# Project20: maternity care mechanisms that improve access and engagement for women with social risk factors in the UK – a mixed-methods, realist evaluation

Hannah Rayment-Jones ![ORCID],[1] Kathryn Dalrymple ![ORCID],[1] James M Harris,[2] Angela Harden,[3] Elidh Parslow,[4] Thomas Georgi,[5] Jane Sandall[1]

[1]Women and Children's Health, King's College London, London, UK
[2]Women's Health, Chelsea And Westminster NHS Foundation Trust, London, UK
[3]Department of Health Services Research and Management, City University of London, London, UK
[4]Maternity and Women's Health, North Middlesex University Hospital NHS Trust, London, UK
[5]Department of Women and Children's Health, King's College London, London, UK

**Correspondence to**
Dr Hannah Rayment-Jones;
hannah.rayment-jones@kcl.ac.uk

## ABSTRACT

**Objectives** To evaluate how women access and engage with different models of maternity care, whether specialist models improve access and engagement for women with social risk factors, and if so, how?

**Design** Realist evaluation.

**Setting** Two UK maternity service providers.

**Participants** Women accessing maternity services in 2019 (n=1020).

**Methods** Prospective observational cohort with multinomial regression analysis to compare measures of access and engagement between models and place of antenatal care. Realist informed, longitudinal interviews with women accessing specialist models of care were analysed to identify mechanisms.

**Main outcome measures** Measures of access and engagement, healthcare-seeking experiences.

**Results** The number of social risk factors women were experiencing increased with deprivation score, with the most deprived more likely to receive a specialist model that provided continuity of care. Women attending hospital-based antenatal care were more likely to access maternity care late (risk ratio (RR) 2.51, 95% CI 1.33 to 4.70), less likely to have the recommended number of antenatal appointments (RR 0.61, 95% CI 0.38 to 0.99) and more likely to have over 15 appointments (RR 4.90, 95% CI 2.50 to 9.61) compared with community-based care. Women accessing standard care (RR 0.02, 95% CI 0.00 to 0.11) and black women (RR 0.02, 95% CI 0.00 to 0.11) were less likely to have appointments with a known healthcare professional compared with the specialist model. Qualitative data revealed mechanisms for improved access and engagement including self-referral, relational continuity with a small team of midwives, flexibility and situating services within deprived community settings.

**Conclusion** Inequalities in access and engagement with maternity care appears to have been mitigated by the community-based specialist model that provided continuity of care. The findings enabled the refinement of a realist programme theory to inform those developing maternity services in line with current policy.

## STRENGTHS AND LIMITATIONS OF THIS STUDY

⇒ The specialist models of care evaluated in this study are situated in the UK's complex maternity system and can be appropriately investigated using a mixed-methods, realist evaluation design.

⇒ Both qualitative and quantitative methods are used to understand not only if access and engagement are improved by specialist models of care, but how and in what context.

⇒ Longitudinal interviews were undertaken to increase trust between participants and the researcher and lessen the perception of surveillance.

⇒ 'The relatively small and varied numbers in each quantitative data group, and the multiple testing required to establish the separate effects of the model of care, place of care and service attended might result in a change in statistical power, reducing the probability of detecting effects when they do exist.'

⇒ The generalisability of the findings is limited by the urban location of both specialist models of care evaluated and the UK health system context.

measured by the timing of antenatal care access and the number of appointments attended.[1] The routine functions of antenatal care include support, health promotion, screening and diagnosis, disease prevention and additional care for women at higher risk.[2] When these functions are of high quality and care is well attended, it makes a crucially important contribution to the reduction of health inequalities.[3 4] The WHO[1] recommends a minimum of eight antenatal appointments to reduce perinatal mortality and improve women's experiences. In high-income countries such as the UK, antenatal care coverage is consistently high and correlates with relatively low maternal and infant mortality rates when compared with the low-income and middle-income countries.[5] Despite this, there are marked inequalities in access, health outcomes and women's

## INTRODUCTION

As a core component of maternity care across the globe, the adequacy of antenatal care is

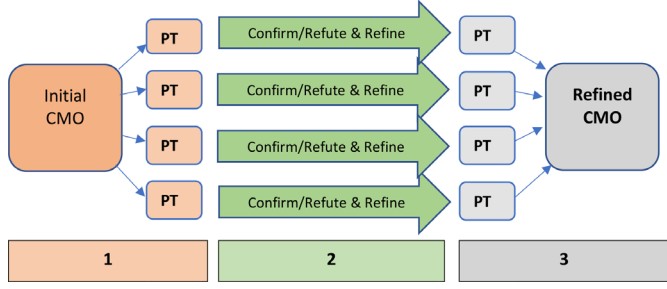

**Figure 1** Programme theory (PT) refinement process leading to context+mechanism=outcome (CMO) configuration.

experiences with poor antenatal care uptake for women from lower socioeconomic and minority groups.[6–10] The most deprived women in the UK are 60% less likely to have received any antenatal care when compared with the least deprived women.[6]

Factors associated with inequalities in maternal and neonatal outcomes include black and minority ethnicity, poverty, young motherhood, homelessness, difficulty speaking or understanding English, migrant or refugee status, domestic violence, mental illness and substance abuse.[7 11–13] It is hypothesised that a lack of access and engagement with maternity services is directly linked to health inequalities for women with these social risk factors.[14–17] Therefore, policies to tackle inequalities often focus on improving access to maternity services and continuity of midwifery care.[18 19] There is a strong evidence base documenting improved maternity outcomes and experiences for women who receive continuity models of midwifery care[20] but less is known about what works to improve access and engagement for women with social risk factors.[21] The impact of place-based maternity care is also poorly understood. The specialist models of care being evaluated in this study provide continuity of care across the antenatal, intrapartum and postnatal periods from a one-to-one midwife or small team of midwives whom women have had the opportunity to meet during pregnancy. One model is based in the community setting and provides care to women in the local, deprived catchment area, the other is based in a large teaching hospital and women with significant social risk factors are referred to the model of care early in pregnancy. See online supplemental file 1 for full definitions of each type of maternity care women may experience, and those being compared in this study. As part of a wider evaluation these two UK-based specialist models, this study aimed to describe and compare service use for women with and without social risk factors by asking:

1. Are women with social risk factors more likely to be offered the specialist models of care over those without? If so, do they find this acceptable?
2. Compared with standard maternity care and group practice models, do specialist models improve access and engagement with maternity services? If so, for whom, in what context and how?

3. Do specialist models of care increase the quality of relational continuity and reduce the number of appointments women miss?
4. How does the location of a model of care-community or hospital, impact on women's access and engagement with maternity services?

## METHODS
### Study design
Realist methodology is a theoretically-informed approach to evaluating complex interventions and often uses mixed methods to understand *how* interventions, in this case models of maternity care, do or do not work in different contexts.[22] This allows those implementing services such as specialist models of care to refine, scale-up or even withdraw the service.[23] The aims of this study were approached through the testing and refinement of initial programme theories constructed in an earlier synthesis of literature[7] and focus groups with midwives[24] relating to how women with social risk factors access and engage with maternity care (see figure 1 for the theory refinement process). Retroductive theorising was used to uncover meaningful causal mechanisms, often focusing on how the wider context and human response to different aspects of maternity care leads to specific outcomes.[25] This approach offers an epistemologically and ontologically grounded way of integrating mixed methods, often analysing qualitative data to find the causal relationship behind quantitative findings.[26] Routinely collected pregnancy and birth outcome data for 1000 women accessing different maternity models, including standard care, group practice and specialist models of care at two large, inner-city maternity services were prospectively collected and analysed. Longitudinal interviews with 20 women with social risk factors were conducted to uncover the mechanisms for any associations found in the cohort analysis. Online supplemental file 1 provides definitions of the two service provider settings and describes the different models of care women might experience at each. Online supplemental file 2 demonstrates the detailed, methodological process of programme theory refinement and development of the context+mechanism=outcome (CMO) configuration and provides adjusted and unadjusted data.

All participants provided written consent by signing a consent form approved by the ethics committee. The quantitative data collected were anonymised before being made available to the research team. A service user panel contributed to the study through external peer review for funding, the research focus, design and analysis of data. The Standards for Quality Improvement Reporting Excellence checklist was used to report this new knowledge for use in healthcare services.[27]

### Patient and public involvement
Multiple representative, diverse groups of service users were involved in the planning and development of this

research. They were recruited through local community groups, clinicians and existing patient involvement groups. Using participatory appraisal methods and online engagement events, recent maternity service users provided feedback on the protocol, study materials, interview guides and refinement of programme theories. They also prioritised outcome measures and reviewed the qualitative data analysis. Training needs were identified by the service users for analysis of quantitative data and further research addressing maternal health inequities.[28]

### Data collection
#### Quantitative data collection
A power calculation was based on a previous analysis of UK antenatal care usage[6] and validated metrics for monitoring local inequalities in access to care at a service evaluated in this research.[29] We calculated that with 250 women in each group (those receiving standard maternity care and those receiving group practice or a specialist model of care), we would have 90% power to detect a 15% difference in timely access to antenatal care (before 12+6 weeks gestation) between the different models of care with 500 anonymised birth records accessed at each trust. Pregnancy and birth outcome data were extracted from computerised records at each service provider. Data collection was prospective; the demographics of the first 500 women booking for maternity care with each service provider in January 2019 were collected, with pregnancy and birth outcomes collected later in the year when all women had been discharged from maternity care. Deprivation deciles calculated using the 2019 English Indices of Deprivation[30] were grouped into four groups of sufficient numbers to enable comparisons between groups of similar numbers.

Semi-structured, longitudinal interviews with 20 women with low socioeconomic status and social risk factors who were receiving specialist care from one of the two service providers were carried out at approximately 28 and 36 weeks' gestation, and 6 weeks post birth. Women were identified by the specialist model midwives providing their care if they met the following inclusion criteria: A deprivation score[31] of higher than 30 and/or secondary school as the highest level of education attained. An interview guide was developed using previous literature[7 24 32] to elicit specific mechanisms explaining how the specialist model might improve access and engagement.

### Analysis
The quantitative data were analysed using Stata V.16.0. First, women's social risk factors, ethnicity, socioeconomic status and medical characteristics were described using descriptive statistics and stratified by the service provider attended to enable comparisons of differences in the samples between each service. Variables were tested for bivariate association using $\chi^2$ tests and t-tests, for dichotomous and continuous variables, respectively. $X^2$ analyses were also performed to test for associations between socioeconomic position by deprivation (indices

of multiple deprivation (IMD)) decile,[33] as well as social and medical risk factors. Second, three regression models were developed: Model 1 adjusted for ethnicity, age, parity, deprivation score, social risk factors and medical risk status; Model 2 included model 1, plus adjustment for the service provider that women attended to consider differences in organisation guidelines, processes and culture; and Model 3 included model 2, however, the place of antenatal care (hospital vs community-based antenatal care) was treated as the independent variable. This structured model allowed us to explore the association between access and engagement depending on the model of care received, while accounting for interactions between independent variables to predict the dependent variable. Risk ratios and CIs are used to demonstrate statistical significance as well as the direction and strength of the effect.[34] Definitions for all demographics and social risk factors are given in online supplemental file 1.

The qualitative data were coded using NVivo V.12 and analysed using a thematic framework analysis.[35] This allowed for the organisation of a large qualitative data set into a coding framework developed using previously constructed programme theories,[7 24] to uncover new theories and differences in women's experiences depending on their characteristics.[35] The method suited the longitudinal approach to data collection as changes in women's perceptions and relationships with healthcare providers could be seen over the course of their pregnancy and postnatal period. We used existing models of data adequacy[36] to assess acceptable data quality. The initial programme theories relating to access and engagement were refined and constructed as a CMO configurations that present specific aspects of the specialist models of care.

## RESULTS
Full pregnancy and birth data were collected and analysed from 799 women accessing care across the two service providers. A total of 201 sets of birth outcome data were missing due to sample dropout, that is, those women stopped receiving care at the service and were therefore excluded from the final analysis. The total number and reasons given for women with missing outcome data at each hospital did not differ significantly. See figure 2 for the data collection flowchart.

### Demographics of women included in the quantitative data analysis
Table 1 presents the demographic profiles of the 799 women who continued their pregnancies and gave birth at the two services, with significant differences in ethnicity, social and medical risk factors, and model of care received highlighted and adjusted for in the analysis of access and engagement outcomes. See online supplemental file 1 for definitions relating to risk factors and model of care.

### Characteristics of the women interviewed
Twenty pregnant women with low socioeconomic status and/or educational attainment were recruited (see

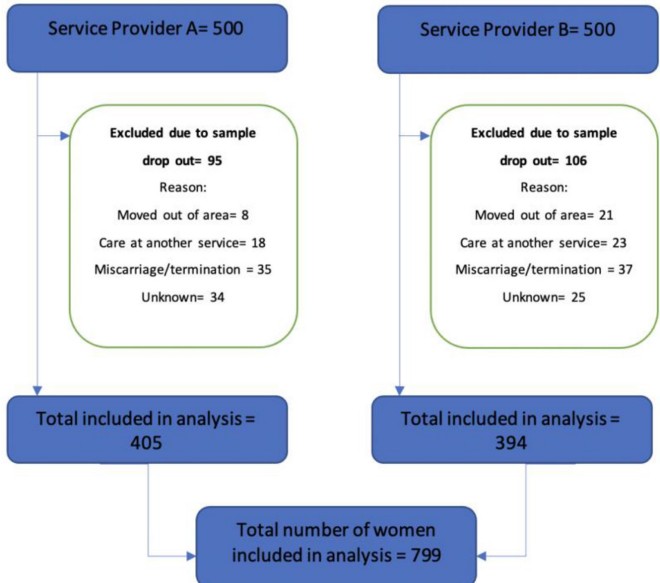

**Figure 2** Quantitative data collection flowchart.

table 2). All 20 women received a specialist model of care, 8 were first time mothers and the other 12 had between one and eight children. For five of the multiparous women, this was their first pregnancy in the UK. Based on 2019 deprivation scores,[30] 19 participants were in the first or second most deprived deciles, with 1 participant in the third and fourth decile group. Twelve participants were born outside of the UK, and nine did not speak English and required an interpreter. All participants were experiencing between one and seven social risk factors including common or severe mental health issues, domestic violence, drug/alcohol misuse, no support, single motherhood, financial and housing problems, learning disability, sexual abuse, trafficking, female genital mutilation and no recourse to public funds. Five participants were seeking asylum, had refugee status or had had an asylum claim refused and nine had social care involvement during their pregnancy.

### Model of care accessed by deprivation score and risk factors
When merging data for the two service providers table 3 shows that women in the most deprived deciles were significantly more likely to receive a specialist model of care, and women in the least deprived deciles were less likely to receive community-based antenatal care than hospital-based antenatal care. This reflects the aim of the service providers to offer women in more deprived areas both specialist and/or community-based antenatal care. A statistically significant relationship was found between deprivation score and the number of social risk factors recorded.

### Testing using qualitative data
Women reported different pathways into the specialist models including direct referrals from a general practitioner (GP), midwives, sexual health clinics, social workers, emergency departments and self-referral. Overall, women were pleased to be referred to the specialist model, however they were often not made aware of *why* they had been referred and described a lack of choice.

I don't know if it's because I have these issues, only, or if it's the team or if it's the hospital or, I don't know, or the combination. I have no idea. But I'm super-happy and feeling lucky I ended up like having all this. like how many people get the support we get here? Is it just me because I have mental health?' (community-based model (CBM)10)

### Timing of access to maternity services
#### Quantitative analysis 1—model of care
The quantitative data were analysed to test the hypothesis that the specialist model of care has an impact on the timing of access to maternity care, known as the 'booking appointment'. Table 4 shows that no relationship was found between the model of care received and the gestation at which women attended their booking appointment. The base outcome was set for less than 10 weeks' gestation at the booking appointment to reflect the National Institute for Care Excellence guidance for women with complex social factors.[15] When adjusting for women's characteristics (see online supplemental file 2 for fully adjusted outcome data tables) primiparous women (risk ratio (RR) 1.79, 95% CI 1.02 to 3.12), those with social risk factors (RR 1.93, 95% CI 1.03 to 3.62) and those with high medical risk status at the booking appointment (RR 2.49, 95% CI 1.18 to 5.25) were more likely to book for maternity care later than 20 weeks.

#### Quantitative analysis 2—place of antenatal care
Table 5 shows that after adjusting for the model of care received and service attended, women receiving hospital-based antenatal care were significantly more likely to attend their booking appointment after 20 weeks' gestation (RR 2.51, 95% CI 1.33 to 4.70).

#### Qualitative data analysis
The qualitative data were analysed to explore why timing of access to maternity care was different for women attending hospital-based antenatal care, primiparous women and those with high medical risk status and social risk factors. Many women expressed wanting to be seen earlier in pregnancy and described not feeling valued by maternity services until their pregnancy is viable. Delays in accessing maternity services was often due to difficulties in accessing GP services and a convoluted referral pathway between community and hospital services. Community-based care and the ability to self-refer to maternity services appeared to reduce the time women spent waiting for their booking appointment.

[at 3 weeks gestation] I called the GP to book an appointment. to say that I'm pregnant, and they sent me… to buy a Clearblue to check it because the list is too long. But it take a very long time, to see the GP… to refer me to the midwife, I was waiting a long time…

**Table 1** Women's demographics at each service provider

| Demographic variable | Service A, n (%) Total data=405 | Service B, n (%) Total data=394 | Total, n (%) Total data=799 | $X^2$, p value |
|---|---|---|---|---|
| Ethnicity | | | | p<0.001 |
| Asian | 37 (9) | 53 (13) | 90 (11) | |
| Black African | 31 (8) | 46 (12) | 77 (10) | |
| Black Caribbean | 23 (6) | 16 (4) | 39 (5) | |
| Black other | 8 (2) | 14 (4) | 22 (3) | |
| Mixed | 12 (3) | 7 (2) | 19 (2) | |
| White British | 98 (24) | 58 (15) | 156 (20) | |
| White other | 80 (20) | 139 (36) | 219 (27) | |
| Unknown | 116 (29) | 61 (15) | 177 (22) | |
| Age | | | | p=0.356 |
| ≤20 | 6 (1) | 4 (1) | 10 (1) | |
| 21–24 years | 19 (5) | 32 (8) | 51 (6) | |
| 25–29 years | 63 (16) | 56 (14) | 119 (15) | |
| 30–34 years | 134 (33) | 125 (32) | 259 (32) | |
| ≥35 years | 183 (45) | 177 (45) | 360 (45) | |
| Parity | | | | p=0.167 |
| Primiparous | 212 (52) | 187 (47) | 399 (50) | |
| Multiparous | 193 (48) | 207 (53) | 400 (50) | |
| Medical risk | | | | |
| High at booking | 118 (29) | 106 (27) | 224 (28) | p=0.496 |
| High at onset of labour | 152 (38) | 223 (57) | 375 (47) | p<0.001 |
| Social risk factor | | | | |
| Domestic abuse | 23 (6) | 17 (4) | 40 (5) | p=0.377 |
| Common mental health | 4 (1) | 34 (9) | 38 (5) | p<0.001 |
| Severe mental health | 2 (<1) | 8 (2) | 10 (1) | p=0.051 |
| Non-English speaking | 16 (4) | 48 (13) | 64 (8) | p<0.001 |
| Social care involvement | 27 (7) | 29 (7) | 56 (7) | p=0.701 |
| Drug/alcohol abuse | 1 (<1) | 10 (3) | 11 (1) | p<0.001 |
| Unsupported/single | 1 (<1) | 11 (3) | 12 (2) | p<0.001 |
| Financial/housing | 15 (4) | 31 (8) | 46 (6) | p<0.001 |
| Learning disability | 6 (2) | 5 (1) | 11 (1) | p=0.797 |
| Sexual abuse/trafficked | 4 (2) | 5 (1) | 9 (1) | p=0.677 |
| Asylum seeker/refugee | 8 (2) | 7 (2) | 15 (2) | p=0.836 |
| Female genital mutilation (FGM) | 0 | 11 (3) | 11 (1) | p<0.001 |
| No recourse to public funds | 6 (1) | 0 | 6 (1) | p<0.001 |
| No of social risk factors | | | | p<0.001 |
| None | 337 (83) | 279 (70) | 616 (77) | |
| 1 | 43 (11) | 61 (15) | 104 (13) | |
| 2 | 13 (3) | 26 (7) | 39 (5) | |
| 3 | 6 (1) | 15 (4) | 21 (3) | |
| 4 | 5 (1) | 9 (2) | 14 (2) | |
| ≥5 | 1 (<1) | 4 (1) | 5 (1) | |
| Name of model of care | | | | p<0.001 |
| Standard care | 256 (63) | 213 (54) | 469 (59) | |
| Group practice | 77 (19) | 144 (37) | 221 (28) | |
| Specialist | 59 (15) | 21 (5) | 80 (10) | |
| Private care | 13 (3) | 16 (4) | 29 (4) | |

Continued

**Table 1**  Continued

| Demographic variable | Service A, n (%) Total data=405 | Service B, n (%) Total data=394 | Total, n (%) Total data=799 | X$^2$, p value |
|---|---|---|---|---|
| Place of model of antenatal care | | | | p<0.001 |
| Standard model in hospital | 100 (25) | 212 (54) | 312 (39) | |
| Standard model in community | 156 (40) | 1 (0) | 157 (20) | |
| Group practice in community | 40 (10) | 94 (24) | 134 (17) | |
| Group practice in hospital | 37 (9) | 50 (13) | 87 (11) | |
| Specialist model in community | 59 (15) | 2 (1) | 61 (8) | |
| Specialist model in hospital | 0 | 19 (5) | 19 (2) | |
| Private care | 13 (3) | 16 (4) | 29 (4) | |
| By place of antenatal care only* | | | | p<0.001 |
| Hospital-based | 137 (35) | 281 (74) | 418 (54) | |
| Community-based | 255 (65) | 97 (26) | 352 (46) | |

*Including all models of care

when I saw the midwife the first time I was around 20 weeks. (CBM7)

### Engagement—number of antenatal appointments attended
#### Quantitative analysis 1—model of care
Women's engagement with services was tested through the number of antenatal appointments women attended. Table 6 shows no significant relationship between the model of care received and the number of antenatal appointments attended. Women with social risk factors were significantly more likely to have more than 15 antenatal appointments than those with no social risk factors (RR 2.57, 95% CI 1.30–5.07), as were women with high medical risk (RR 2.70, 95% CI 1.21 to 6.03) (see online supplemental file 2).

#### Quantitative analysis 2—place of antenatal care
Table 7 shows that women receiving hospital-based care were significantly likely to have less than the recommended number of appointments,[1 2 15] (RR 0.61, 95% CI 0.38 to 0.99), and much more likely to have over 15 appointments (RR 4.90, 95% CI 2.50 to 9.61) than those receiving community-based care regardless of their medical risk status and other confounding factors.

#### Qualitative analysis
Flexible care was discussed by women accessing both specialist models, in terms of when and where appointments were scheduled and how long each appointment lasted. The community or home setting was seen as more convenient and supportive, particularly for those who were unfamiliar with UK transport systems, and those with little resources or young children. Women from both specialist models expressed ease of contacting the specialist model midwives through phone call, text messaging or email, and felt that it reduced the number of face-to-face appointments they needed and that they could rely on the midwives to remind them of appointments. Women who did not speak fluent English (45%)

particularly valued the ability to text their midwife as it gave them the opportunity to use translation technology. This flexibility appeared to encourage women to seek help more readily when they were concerned or needed reassurance. However, women in the hospital-based model did not always feel that the way care was scheduled suited their needs but structured around the hospital protocol and organisational efficiency.

If it's [antenatal care] like near to me it's OK but… I have so much financial problems so when she [HBM midwife] came to my house like, so much easier for me….and at the beginning I didn't know how to use bus… she helped me. (hospital-based model (HBM)6)

### The quality of relational continuity and missed appointments
#### Quantitative analysis 1—model of care
The quantitative data were analysed to test the hypothesis that the specialist model of care increases the quality of relational continuity through the number of antenatal appointments and care in labour from a known healthcare professional (midwife, GP or obstetrician), and reduces the number of appointments women miss, or do not attend. Table 8 shows that women receiving group practice and specialist models were significantly more likely to see a known healthcare professional during pregnancy. When adjusting the model for women's characteristics black African women were the least likely group to see a known healthcare professional more than five times (RR 0.17, 95% CI 0.03 to 0.82). Women receiving care in the specialist model were significantly more likely to be looked after in labour by a known healthcare professional than the group practice (RR 0.44, 95% CI 0.21 to 0.92).

#### Quantitative analysis 2—place of antenatal care
Table 9 shows that women receiving hospital-based antenatal care were less likely to see a known healthcare professional for their antenatal appointments compared

**Table 2** Demographics of women interviewed

| Characteristic | Community-based model n=10 | Hospital-based model n=10 | Total n (%) n=20 |
|---|---|---|---|
| Ethnicity and migration status | | | |
| Born outside the UK: | 7 | 5 | 12 (60) |
| Asian | 0 | 2 | 2 (10) |
| Black African | 3 | 0 | 3 (15) |
| Black Caribbean | 0 | 1 | 1 (5) |
| White | 4 | 2 | 6 (30) |
| Asylum seeker/refugee* | 2 | 3 | 5 (25) |
| Born inside the UK: | 3 | 5 | 8 (40) |
| Asian British | 1 | 1 | 2 (10) |
| Black British | 2 | 1 | 3 (15) |
| White British | 0 | 3 | 3 (15) |
| Not proficient in English language/interpreter required | 5 | 4 | 9 (45) |
| Age | | | |
| 18–24 | 0 | 3 | 3 (13) |
| 25–29 | 1 | 1 | 2 (2) |
| 30–34 | 5 | 5 | 10 (50) |
| >34 | 4 | 1 | 5 (25) |
| Parity | | | |
| Primiparous | 5 | 3 | 8 (40) |
| IMD decile (2019) | | | |
| Most deprived 1st+2nd | 9 | 10 | 19 (95) |
| 3rd and 4th | 1 | 0 | 1 (5) |
| Least deprived 5th to 10th | 0 | 0 | 0 |
| No. of social risk factors | | | |
| 1 | 3 | 0 | 3 (15) |
| 2 | 0 | 2 | 2 (10) |
| 3 | 2 | 0 | 2 (10) |
| 4 | 1 | 1 | 2 (10) |
| ≥5 | 4 | 7 | 11 (55) |
| Mental illness | | | |
| Common | 5 | 9 | 14 (70) |
| Severe | 1 | 1 | 2 (15) |
| Level of education | | | |
| Secondary school only | 5 | 6 | 14 (70) |
| Completed college | 4 | 3 | 2 (15) |
| Completed university | 1 | 1 | 4 (20) |
| High medical risk at booking | 7 | 5 | 12 (60) |

*Including women whose asylum claim had been refused.
IMD, indices of multiple deprivation.

with those receiving community-based care. No significant relationship was found between the place of care and the number of women cared for in labour by their named midwife.

## Missed appointments
### Quantitative analysis 1—model of care
Table 10 shows that no significant relationship was found between model of care and the number of missed appointments. When adjusting for women's characteristics,

multiparous women were four times more likely (RR 4.50, 95% CI 1.13 to 17.82) to miss four or more appointments than primiparous women, and the older a woman was less likely that she will miss four or more appointments (RR 0.03, 95% CI 0.00 to 0.55). Black African women (RR 12.85, 95% CI 2.42 to 68.07), and women with social risk factors (RR 2.26, 95% CI 1.14 to 4.47) were more likely to miss two or more appointments (see online supplemental file 2). These findings should be viewed with caution due to the wide CIs.

### Analysis 2—place of antenatal care
Table 11 shows no significant relationship was found between the number of missed appointments and the place of antenatal care.

### Qualitative analysis
Women accessing the community-based model perceived a high level of continuity of care from the whole team. Women accessing the hospital-based model discussed how they knew their named midwife but not necessarily the rest of the team, impacting on how they sought help. They were also not always aware of the level of continuity offered, expressed more anxiety about their labour care and discussed the impact of midwives needing to cancel appointments. For some, cancelled appointments and a lack of continuity had a significant impact on their support and engagement, leading to social care involvement. Women in both models described feeling comfortable when rebooking appointments; they were unable to attend that appeared to reduce the number of 'missed' appointments recorded, seen as a mitigating effect of the specialist model.

> If I'm running late or something comes up I just text her or give her a call and she'll muddle things around. (HBM9)

### Summary of findings
Table 12 summarises the quantitative findings in relation to either the model of care received, or the place of antenatal care, or both. Characteristics of women at disproportionate risk and differences between the services providers is also presented.

### Refined programme theory
The qualitative interviews with women enabled the refinement of previously constructed programme theories,[7 24] giving insight into specific, underlying mechanisms of improved access and engagement with maternity services. The process of refinement using the data analysed is detailed in online supplemental file 2. The CMO configurations are detailed in table 13, providing a framework for those developing future models of care for women with social risk factors.

## DISCUSSION
In response to concerns that the most affluent and lowest risk women are often the most likely to receive the

**Table 3** Maternity care received, and risk factors recorded by deprivation score

| Maternity care and risk factors | Least deprived (7th, 8th, 9th + 10th) | 5th and 6th deciles | 3rd and 4th deciles | Most deprived (1st + 2nd deciles) | Total n (%) | $X^2$ p value |
|---|---|---|---|---|---|---|
| Model of care | Total=149 | Total=158 | Total=286 | Total=206 | N=799 | p<0.001 |
| Standard care | 81 (54) | 98 (62) | 163 (57) | 127 (62) | 469 (58) | |
| Group practice | 43 (29) | 41 (26) | 85 (30) | 52 (25) | 221 (28) | |
| Specialist | 3 (2) | 16 (10) | 34 (12) | 27 (13) | 80 (10) | |
| Private care | 22 (15) | 3 (2) | 4 (1) | 0 (0) | 29 (4) | |
| Place of antenatal care* | Total=127 | Total=155 | Total=282 | Total=206 | Total=770 | p<0.001 |
| Hospital-based | 82 (65) | 91 (59) | 134 (48) | 111 (54) | 418 (54) | |
| Community-based | 45 (35) | 64 (41) | 148 (52) | 95 (46) | 352 (46) | |
| Social risk factors | Total=149 | Total=158 | Total=286 | Total=206 | N=799 | p<0.001 |
| None | 133 (89) | 132 (84) | 212 (74) | 139 (67) | 616 (77) | |
| One | 10 (7) | 17 (11) | 50 (17) | 27 (13) | 104 (13) | |
| Two | 5 (3) | 4 (3) | 10 (4) | 20 (10) | 39 (5) | |
| Three | 1 (1) | 2 (1) | 10 (4) | 8 (4) | 21 (3) | |
| Four or more | 0 | 3 (2) | 4 (1) | 12 (6) | 19 (2) | |
| Medical risk | | | | | | |
| High at booking | 37 (25) | 46 (29) | 85 (30) | 56 (27) | 224 (28) | p=0.743 |
| High at onset of labour | 62 (42) | 79 (50) | 138 (48) | 96 (47) | 375 (47) | p=0.475 |

*Excludes private care.

highest standards of maternity care,[37] a clear example of Tudor Hart's Inverse Care Law,[38] this study examines a shift change to more proportionate universalism through services targeting those who are at highest risk of poor outcomes. The aims of the specialist models of care to reach those who are most at risk of poor outcomes and experiences, are being met. This is an important finding as it addresses the aims of the National Health Service 10-year plan[39] to improve outcomes and care experiences through the offer of continuity of care to black and minority ethnic women and those living in deprived areas. The findings suggest that placing continuity of care models in areas of deprivation are likely to identify women who are experiencing social risk factors during pregnancy, thus providing them with an enhanced level of care whether they have disclosed risk factors or not. In contrast, strict inclusion criteria to hospital-based models risk missing those women who are at increased risk but are

**Table 4** Gestation at booking appointment in relation to the model of care received

| Gestation at booking | Model of care | Number of women (%) | Unadjusted RR (95% CI) | Model 1 Adjusted RR (95% CI)* | Model 2 Adjusted RR (95% CI)† | Model 3 Adjusted RR (95% CI)‡ |
|---|---|---|---|---|---|---|
| <10 weeks | Standard | 197 (62) | Ref | Ref | Ref | Ref |
| | Group | 87 (27) | Ref | Ref | Ref | Ref |
| | Specialist | 35 (11) | Ref | Ref | Ref | Ref |
| 10–12 | Standard | 166 (58) | 1.01 (0.59 to 1.73) | 0.91 (0.52 to 1.61) | 0.77 (0.42 to 1.40) | 0.76 (0.42 to 1.38) |
| | Group | 90 (32) | 1.24 (0.70 to 2.12) | 1.18 (0.65 to 2.16) | 1.09 (0.59 to 2.02) | 0.86 (0.46 to 1.63) |
| | Specialist | 29 (30) | Ref | Ref | Ref | Ref |
| 13–20 | Standard | 53 (61) | 0.94 (0.43 to 2.02) | 1.14 (0.50 to 2.61) | 0.88 (0.37 to 2.09) | 0.89 (0.37 to 2.13) |
| | Group | 24 (28) | 0.96 (0.41 to 2.22) | 1.14 (0.47 to 2.77) | 1.04 (0.42 to 2.56) | 0.75 (0.29 to 1.94) |
| | Specialist | 10 (11 | Ref | Ref | Ref | Ref |
| >20 weeks | Standard | 53 (68) | 1.56 (0.62 to 3.92) | 2.19 (0.82 to 5.80) | 1.29 (0.46 to 3.60) | 1.27 (0.45 to 3.56) |
| | Group | 19 (24) | 1.27 (0.46 to 3.45) | 1.75 (0.61 to 4.99) | 1.29 (0.43 to 3.82) | 1.15 (0.38 to 3.44) |
| | Specialist | 6 (8) | Ref | Ref | Ref | Ref |

*Model 1: Adjusted for demographics ethnicity, age, parity, IMD score, any social and medical risk factors at booking and onset of labour.
†Model 2: Model 1 + adjustment for place of antenatal care (community or hospital).
‡Model 3: Model 2 + adjustment for service provider attended (A or B).
IMD, indices of multiple deprivation; RR, risk ratio.

**Table 5** Gestation at booking appointment in relation to the place of antenatal care

| Gestation at booking | Place of antenatal care | Number of women (%) | Unadjusted RR | Model 1 Adjusted RR (95% CI)* | Model 2 Adjusted RR (95% CI)† | Model 3 Adjusted RR (95% CI)‡ |
|---|---|---|---|---|---|---|
| <10 | Hospital | 154 (48) | Ref | Ref | Ref | Ref |
|  | Community | 165 (52) | Ref | Ref | Ref | Ref |
| 10–13 weeks | Hospital | 154 (54) | 1.25 (0.91–1.73) | 1.19 (0.85 to 1.66) | 1.30 (0.91 to 1.86) | 1.00 (0.67 to 1.49) |
|  | Community | 131 (46) | Ref | Ref | Ref | Ref |
| 13–20 | Hospital | 54 (62) | 1.75 (1.07–2.84) | 1.59 (0.95 to 2.66) | 1.66 (0.97 to 2.87) | 1.05 (0.54 to 2.01) |
|  | Community | 33 (38) | Ref | Ref | Ref | Ref |
| >20 weeks | Hospital | 56 (72) | 2.72 (1.58–4.67) | 2.89 (1.65 to 5.10) | 2.81 (1.56 to 5.06) | 2.51 (1.33 to 4.70) |
|  | Community | 22 (28) | Ref | Ref | Ref | Ref |

*Model 1: Adjusted for demographics ethnicity, age, parity, IMD score, social risk, and medical risk factors at booking and onset of labour.
†Model 2: Model 1 + adjusted for model of care.
‡Model 3: Model 2 + adjusted for service provider attended.
IMD, indices of multiple deprivation; RR, risk ratio.

yet to disclose social risk factors, leading to fragmented care and further risk of 'falling through the gaps'. The wider Project20 evaluation tests the theory that women will feel more able to disclose risk factors when they have developed a trusting relationship with their care provider (paper currently under review; WOMBI-D-22–00309 R2)

Overall women found the referral to both specialist models of care acceptable, but some expressed a lack of choice, and perceived discrimination about being left in the dark for the reason for referral. There is a wealth of literature around healthcare professionals' reluctance to address sensitive issues such as mental health, social risk factors and cultural differences with women.[40–47] In addition to this, socioeconomic status and ethnicity are often associated with stigma and discrimination in healthcare services.[48–51] These factors might contribute to midwives' feelings of discomfort around informing women why

they have been referred to a specialist model. Perceived discrimination could be avoided through community-based care within areas of deprivation rather than inclusion criteria based on these social risk factors.

The findings showed no differences between the model of care received and timing of access to maternity care or the number of antenatal appointments women attended. Considering women in the specialist models were more likely to have low socioeconomic status and social risk factors and therefore more likely to struggle to access and engage.[6 52 53] This indicates that the inequality in access may have been mitigated by the specialist model of care. This theory would need to be tested with a larger sample. Regardless of the model of care women received, those receiving hospital-based care were more likely to attend their first maternity appointment after 20 weeks' gestation, this was thought to be partly due to convoluted

**Table 6** Number of antenatal appointments attended in relation to the model of care accessed

| Number of antenatal appointments | Model of care | Number of women (%) | Unadjusted RR | Model 1 Adjusted RR (95% CI)* | Model 2 Adjusted RR (95% CI)† | Model 3 Adjusted RR (95% CI)‡ |
|---|---|---|---|---|---|---|
| 1–6 | Standard | 173 (63) | 0.58 (0.31–1.09) | 0.64 (0.35 to 1.30) | 0.86 (0.44 to 1.69) | 0.87 (0.44 to 1.70) |
|  | Group | 63 (23) | 0.41 (0.21–0.80) | 0.42 (0.22 to 0.93) | 0.43 (0.21 to 0.86) | 0.55 (0.25 to 1.15) |
|  | Specialist | 37 (14) | Ref | Ref | Ref | Ref |
| 7–9§ | Standard | 135 (61) | Ref | Ref | Ref | Ref |
|  | Group | 70 (31) | Ref | Ref | Ref | Ref |
|  | Specialist | 17 (8) | Ref | Ref | Ref | Ref |
| 10–14 | Standard | 120 (64) | 0.94 (0.45–1.95) | 1.13 (0.53 to 2.42) | 0.80 (0.36 to 1.78) | 0.82 (0.37 to 1.81) |
|  | Group | 52 (28) | 0.78 (0.36–1.70) | 0.90 (0.40 to 2.02) | 0.82 (0.36 to 1.86) | 0.97 (0.42 to 2.24) |
|  | Specialist | 16 (8) | Ref | Ref | Ref | Ref |
| ≥15 | Standard | 38 (46) | 0.47 (0.20–1.13) | 0.68 (0.26 to 1.74) | 0.43 (0.16 to 1.17) | 0.43 (0.16 to 1.14) |
|  | Group | 35 (42) | 0.85 (0.35–2.05) | 1.25 (0.47 to 3.28) | 1.02 (0.38 to 2.75) | 1.28 (0.47 to 3.45) |
|  | Specialist | 10 (12 | Ref | Ref | Ref | Ref |

*Model 1: Adjusted for demographics ethnicity, age, parity, IMD score, any social and medical risk factors at booking and onset of labour.
†Model 2: Model 1 + adjustment for place of antenatal care (community or hospital).
‡Model 3: Model 2 + adjustment for service provider attended (A or B).
§Set as base as WHO recommends eight antenatal appointments.[1]
IMD, indices of multiple deprivation; RR, risk ratio.

**Table 7** Number of antenatal appointments attended in relation to the place of antenatal care

| Number of antenatal appointments | Model of care | Number of women (%) | Unadjusted RR | Model 1 Adjusted RR (95% CI)* | Model 2 Adjusted RR (95% CI)† | Model 3 Adjusted RR (95% CI)‡ |
|---|---|---|---|---|---|---|
| 1–6 | Hospital | 106 (39) | 0.53 (0.37–0.77) | 0.56 (0.38 to 0.82) | 0.47 (0.31 to 0.71) | 0.61 (0.38 to 0.99) |
| | Community | 167 (61) | Ref | Ref | Ref | Ref |
| 7–9§ | Hospital | 120 (54) | Ref | Ref | Ref | Ref |
| | Community | 102 (46) | Ref | Ref | Ref | Ref |
| 10–14 | Hospital | 130 (69) | 1.90 (1.26–2.86) | 2.22 (1.44 to 3.41) | 2.27 (1.43 to 3.58) | 2.70 (1.62 to 4.49) |
| | Community | 58 (31) | Ref | Ref | Ref | Ref |
| ≥15 | Hospital | 60 (72) | 2.21 (1.28–3.83) | 2.92 (1.59 to 5.34) | 3.36 (1.80 to 6.25) | 4.90 (2.50 to 9.61) |
| | Community | 23 (28) | Ref | Ref | Ref | Ref |

*Model 1: Adjusted for demographics ethnicity, age, parity, IMD score, social risk, and medical risk factors at booking and onset of labour.
†Model 2: Model 1 + adjusted for model of care.
‡Model 3: Model 2 + adjusted for service provider attended.
§Set as base as WHO recommends eight antenatal appointments.[1]
IMD, indices of multiple deprivation; RR, risk ratio.

referral systems between the community and hospital. Many women expressed wanting to be seen earlier in pregnancy, challenging the notion that women with social risk factors do not prioritise their maternity care.[54] Women also described difficulties in registering or booking an appointment with their GP when they found out they were pregnant, particularly if they did not speak English, were homeless or unfamiliar with the system. The wider literature has identified similar barriers to access including difficulties navigating the health system and service delays in the processing of referrals.[52 55 56] A UK study[52] of women accessing antenatal care in a multicultural,

**Table 8** Number of appointments and support in labour by known healthcare professional

| Number of antenatal appointments with a known professional | Model of care | Number of women (%) | Unadjusted OR | Model 1 Adjusted RR (95% CI)* | Model 2 Adjusted RR (95% CI)† | Model 3 Adjusted RR (95% CI)‡ |
|---|---|---|---|---|---|---|
| None | Standard | 313 (69) | Ref | Ref | Ref | Ref |
| | Group | 84 (19) | Ref | Ref | Ref | Ref |
| | Specialist | 53 (12) | Ref | Ref | Ref | Ref |
| 1–2 | Standard | 108 (73) | 6.09 (1.86–19.9) | 4.43 (1.30 to 15.0) | 2.75 (0.78 to 9.67) | 2.12 (0.57 to 7.87) |
| | Group | 36 (24) | 7.57 (2.21–25.8) | 7.30 (2.05 to 26.0) | 6.81 (1.89 to 24.5) | 1.81 (0.44 to 7.45) |
| | Specialist | 3 (2) | Ref | Ref | Ref | Ref |
| 3 | Standard | 36 (47) | 0.76 (0.33–1.72) | 0.69 (0.27 to 1.77) | 0.47 (0.17 to 1.29) | 0.34 (0.11 to 1.06) |
| | Group | 32 (42) | 2.52 (1.08–5.89) | 3.33 (1.27 to 8.75) | 3.15 (1.18 to 8.38) | 0.82 (0.24 to 2.75) |
| | Specialist | 8 (11) | Ref | Ref | Ref | Ref |
| 4 | Standard | 7 (15) | 0.29 (0.83–1.04) | 0.23 (0.05 to 0.89) | 0.16 (0.03 to 0.67) | 0.10 (0.02 to 0.53) |
| | Group | 34 (76) | 5.36 (1.80–15.9) | 5.99 (1.78 to 20.1) | 5.19 (1.62 to 18.7) | 1.72 (0.39 to 1.47) |
| | Specialist | 4 (9) | Ref | Ref | Ref | Ref |
| >5 | Standard | 5 (10) | 0.76 (0.02–0.23) | 0.05 (0.01 to 0.19) | 0.03 (0.00 to 0.13) | 0.02 (0.00 to 0.11) |
| | Group | 35 (68) | 2.00 (0.93–4.29) | 2.50 (0.96 to 6.49) | 2.38 (0.90 to 6.32) | 0.82 (0.23 to 2.94) |
| | Specialist | 11 (22) | Ref | Ref | Ref | Ref |
| Looked after in labour by a known midwife | Standard | 235 (63) | 0.41 (0.24–0.71) | 0.44 (0.24 to 0.82) | 0.62 (0.32 to 1.19) | 0.59 (0.30 to 1.17) |
| | Group | 81 (23) | 0.24 (0.13–0.42) | 0.23 (0.12 to 0.44) | 0.24 (0.12 to 0.47) | 0.44 (0.21 to 0.92) |
| | Specialist | 53 (14) | Ref | Ref | Ref | Ref |

*Model 1: Adjusted for demographics ethnicity, age, parity, IMD score, any social and medical risk factors at booking and onset of labour.
†Model 2: Model 1 + adjustment for place of antenatal care (community or hospital).
‡Model 3: Model 2 + adjustment for service attended (A or B).
IMD, indices of multiple deprivation; RR, risk ratio.

**Table 9** Number of appointments with a known healthcare professional in relation to place of care

| Number of antenatal appointments with a known professional | Model of care | Number of women (%) | Unadjusted OR | Model 1 Adjusted RR (95% CI)* | Model 2 Adjusted RR (95% CI)† | Model 3 Adjusted RR (95% CI)‡ |
|---|---|---|---|---|---|---|
| None | Hospital | 215 (48) | Ref | Ref | Ref | Ref |
|  | Community | 235 (52) | Ref | Ref | Ref | Ref |
| 1–2 | Hospital | 109 (74) | 3.15 (2.08–4.76) | 2.44 (1.56 to 3.83) | 2.71 (1.63 to 4.50) | 0.24 (0.10 to 0.59) |
|  | Community | 38 (26) | Ref | Ref | Ref | Ref |
| 3 | Hospital | 45 (59) | 1.59 (0.97–2.61) | 1.23 (0.71 to 2.13) | 2.25 (1.19 to 4.25) | 0.23 (0.08 to 0.62) |
|  | Community | 31 (41) | Ref | Ref | Ref | Ref |
| 4 | Hospital | 20 (44) | 0.87 (0.47–1.62) | 0.61 (0.31 to 1.22) | 2.02 (0.92 to 1.47) | 0.30 (0.11 to 0.83) |
|  | Community | 25 (56) | Ref | Ref | Ref | Ref |
| >5 | Hospital | 28 (55) | 1.33 (0.74–2.39) | 0.70 (0.35 to 1.39) | 2.66 (1.24 to 5.70) | 0.35 (0.12 to 0.96) |
|  | Community | 23 (45) | Ref | Ref | Ref | Ref |
| Looked after in labour by a known midwife | Hospital | 153 (41) | 0.34 (0.25–0.46) | 0.46 (0.33 to 0.64) | 0.39 (0.27 to 0.57) | 0.89 (0.59 to 1.36) |
|  | Community | 216 (59) | Ref | Ref | Ref | Ref |

*Model 1: Adjusted for demographics ethnicity, age, parity, IMD score, social risk, and medical risk factors at booking and onset of labour.
†Model 2: Model 1 + adjusted for model of care.
‡Model 3: Model 2 + adjusted for service attended.
IMD, indices of multiple deprivation; RR, risk ratio.

deprived area found that women want to access care in early pregnancy but perceive antenatal care for viable and continuing pregnancies at a later gestation. If women do not feel that a service is open to them, or that maternity services only value those who carry a viable pregnancy, they may internalise this as a prioritisation of the well-being of the fetus over their own emotional, physical and social needs. The wider Project20 evaluation previously tested programme theory relating to interpreter services for pregnant women with social risk factors, finding that despite accessing a specialist model of care women experienced a lack of regulation and access to

**Table 10** Number of missed appointments in relation to model of care received

| Number of missed appointments | Model of care | Number of women (%) | Unadjusted OR | Model 1 Adjusted RR (95% CI)* | Model 2 Adjusted RR (95% CI)† | Model 3 Adjusted RR (95% CI)‡ |
|---|---|---|---|---|---|---|
| None | Standard | 352 (63) | Ref | Ref | Ref | Ref |
|  | Group | 147 (26) | Ref | Ref | Ref | Ref |
|  | Specialist | 62 (11) | Ref | Ref | Ref | Ref |
| 1 | Standard | 62 (56) | 1.36 (0.62–2.99) | 1.58 (0.67 to 3.67) | 1.46 (0.61 to 3.48) | 1.44 (0.61 to 3.49) |
|  | Group | 40 (37) | 2.10 (9.33–4.76) | 2.41 (1.01 to 5.77) | 2.38 (0.99 to 5.72) | 1.97 (0.80 to 4.84) |
|  | Specialist | 8 (7) | Ref | Ref | Ref | Ref |
| 2 | Standard | 37 (67) | 1.62 (0.56–4.73) | 2.27 (0.72 to 7.13) | 1.79 (0.54 to 5.84) | 1.81 (0.53 to 6.19) |
|  | Group | 14 (26) | 1.47 (0.46–4.66) | 1.83 (0.54 to 6.20) | 1.79 (0.52 to 6.14) | 0.86 (0.22 to 3.33) |
|  | Specialist | 4 (7) | Ref | Ref | Ref | Ref |
| 3 | Standard | 14 (52) | 1.23 (0.27–5.55) | 2.71 (0.33 to 22.0) | 2.01 (0.23 to 17.2) | 2.02 (0.22 to 18.4) |
|  | Group | 11 (41) | 2.31 (0.49–10.7) | 4.54 (0.54 to 37.9) | 4.41 (0.51 to 37.6) | 2.03 (0.25 to 23.6) |
|  | Specialist | 2 (7) | Ref | Ref | Ref | Ref |
| ≥4 | Standard | 4 (24) | 0.17 (0.42–0.72) | 0.20 (0.04 to 0.98) | 0.22 (0.04 to 1.14) | 0.23 (0.04 to 1.22) |
|  | Group | 9 (53) | 0.94 (0.28–3.19) | 0.94 (0.22 to 4.02) | 0.94 (0.21 to 4.05) | 0.49 (0.08 to 2.71) |
|  | Specialist | 4 (23) | Ref | Ref | Ref | Ref |

*Model 1: Adjusted for demographics ethnicity, age, parity, IMD score, any social and medical risk factors at booking and onset of labour.
†Model 2: Model 1 + adjustment for place of antenatal care (community or hospital).
‡Model 3: Model 2 + adjustment for service provider attended (A or B).
IMD, indices of multiple deprivation; RR, risk ratio.

**Table 11** Number of missed appointments in relation to place of care

| Number of missed appointments | Place of care | Number of women (%) | Unadjusted OR | Model 1 Adjusted RR (95% CI)* | Model 2 Adjusted RR (95% CI)† | Model 3 Adjusted RR (95% CI)‡ |
|---|---|---|---|---|---|---|
| None | Hospital | 292 (52) | Ref | Ref | Ref | Ref |
| | Community | 269 (48) | Ref | Ref | Ref | Ref |
| 1 | Hospital | 64 (58) | 1.28 (0.84–1.93) | 1.11 (0.71 to 1.72) | 1.21 (0.76 to 1.94) | 0.95 (0.56 to 1.62) |
| | Community | 46 (42) | Ref | Ref | Ref | Ref |
| 2 | Hospital | 38 (69) | 2.06 (1.13–3.74) | 2.07 (1.12 to 3.88) | 1.96 (1.01 to 3.78) | 0.64 (0.26 to 1.58) |
| | Community | 17 (31) | Ref | Ref | Ref | Ref |
| 3 | Hospital | 18 (67) | 1.84 (0.81–4.17) | 1.82 (0.74 to 4.46) | 2.18 (0.83 to 5.67) | 1.02 (0.32 to 3.20) |
| | Community | 9 (33) | Ref | Ref | Ref | Ref |
| ≥4 | Hospital | 6 (35) | 0.50 (0.18–1.38) | 0.48 (0.15 to 1.48) | 0.77 (0.23 to 2.51) | 0.36 (0.08 to 1.55) |
| | Community | 11 (65) | Ref | Ref | Ref | Ref |

*Model 1: Adjusted for demographics ethnicity, age, parity, IMD score, social risk, and medical risk factors at booking and onset of labour.
†Model 2: Model 1 + adjusted for model of care.
‡Model 3: Model 2 + adjusted for service provider attended.
IMD, indices of multiple deprivation; RR, risk ratio.

high-quality interpretation services.[57] As well as impacting on women's engagement with services, these factors can impact on the safety of women with complex needs such as those who are experiencing abuse, poor mental health or need to discuss a termination of pregnancy. Previous research has found a correlation between continuity models of care and increased disclosure and referral to support services,[58 59] but it must be acknowledged that under ascertainment of sensitive issues such as mental health and domestic abuse remains likely while women perceive services as a form of surveillance and risk.[59]

A recent review of the literature on how women with social risk factors experience maternity care in the UK found reasons for late access include the denial of services based on a lack of documentation, fear of disclosure to immigration services, language and financial barriers, cultural differences, unfamiliarity, a lack of trust and a perception that maternity services act as a system of surveillance rather than support.[7] Practice recommendations

detailed in the refined programme theory suggest that all women are made aware of the possibility to self-refer directly to maternity services at the first point of contact with health services, using language appropriate information. Strict inclusion criteria for access to specialist models that restrict women who have booked late for maternity care should be reviewed as these women are often high risk and can benefit from the specialist model regardless of their gestation. An evaluation of group antenatal care[60] included women who had booked later in pregnancy with positive feedback and no negative impact on group dynamics. With the national expansion of continuity of care models,[18] inclusion criteria could be relaxed somewhat as the demands on the service are more evenly distributed. Future research should assess the impact of place-based midwifery continuity of care on health inequalities to test the outcomes of this preference.

This study reflects findings in the wider literature[6 61] highlighting service use inequities for women

**Table 12** Overview of quantitative outcomes

| Access and engagement outcome variable | Characteristics of women at disproportionate risk when adjusting (online supplemental file 2) | Significant effect of specialist model of care | Significant effect of hospital-based antenatal care | Significant effect of service |
|---|---|---|---|---|
| Access to specialist model | Most deprived, social risk factors, black African ethnicity | ↑ | ↓ | = |
| Late gestation at booking appointment (>20/40) | Primiparous, high medical risk, social risk factors | = | ↑ | = |
| Number of antenatal appointments outside of recommendations[1 2] | Social risk factors and high medical risk status (>15 appointments) | = | ↓/↑ | B↓ |
| Appointments with a known healthcare professional | Black African (least likely to see a known healthcare professional) | = | ↓ | B↓ |
| Missed appointments | Multiparous, black African, social risk | = | = | B↑ |

↑=statistically significant increase (p<0.05), ↓=statistically significant decrease (p<0.05), =no significant relationship detected, 'A' and 'B' refer to service providers.

**Table 13** Refined programme theory—access and engagement with maternity services

| Context + | Mechanism = | Outcome |
|---|---|---|
| Women who struggle to access services and are at greater risk of booking for maternity care at a late gestation.<br>Women who are unfamiliar with the UK health system or have difficulties in registering with health services. These women are often experiencing social risk factors that might lead to chaotic lives, social isolation, lack of resource, lack of support.<br>Primiparous women, those with any social risk factor, and high medical risk are more likely to book late for maternity care. Multiparous women, black African women and those with social risk factors are more likely to miss appointments. Black African women have less appointments with a known healthcare professional. | (M1) If maternity care provision commences when a woman accesses services regardless of her gestation and women have 24/7 access to a small team of midwives whom they have had the opportunity to meet during pregnancy and are encouraged to contact via a phone call, text message or free technology. | (O1) Then women would not feel unsupported, anxious and that the service does not value them until they have a viable pregnancy. This might also improve early access to safe abortion and family planning services. Engagement with services will improve through needs-based communication and appropriately timed antenatal appointments. This open access can work both ways through midwives reminding them of appointments, this leads to women feeling more 'cared for'. |
| | (M2) If women are made aware of the possibility and how to self-refer to maternity services and specialist models of care by administrative staff at the first point of contact. | (O2) Then difficulties trying to access a GP will be overcome, the time spent waiting for a GP appointment reduced and long referral processes between primary and secondary services will be avoided. |
| | (M3) If women can access a community-based service where GP's and midwives regularly communicate with each other. | (O3) Then the timing of access to a booking appointment with a midwife will be improved and convoluted referral pathways between community and hospital services avoided. |
| | (M4) If women living in areas of deprivation are prioritised to receive continuity of care through community-based models. | (O4) Then services are likely to provide an enhanced package of care to women with social risk factors who have not previously disclosed these issues with professionals, and care is less likely to be disrupted during pregnancy when a disclosure is made. |
| | (M5) If women are informed of the reasons why they have been allocated a continuity of care model or specialist service, or are able to self-refer to them if they feel they are eligible for their care. | (O5) Then the development of a trusting and open relationship with their healthcare provider will be enabled and feelings of suspicion and surveillance reduced. This transparency may also reduce feelings of discrimination for women in marginalised groups. |
| | (M6) If women who book late for pregnancy care are eligible for specialist models of care where they have a named midwife or small team of midwives. | (O6) Then the benefits of these models of care and highlighted mechanisms may protect them from the disproportionately poor outcomes associated with late booking. |
| | (M7) If women can reschedule appointments easily, and do not fear judgement or reproach when they miss appointments. | (O7) Then they will perceive the maternity environment as a place of safety and their engagement with flexible services will improve. |
| | (M8) If women have the opportunity to get to know all the midwives in a small team throughout their pregnancy and the number, time and place of appointments is co-planned with women to meet their individual needs. | (O8) Then they will not feel disappointment or let down when their named midwife is unable to attend an appointment. Care, information and responsibility will be shared across the team, women will be better able to engage, not perceive the pressure of time and feel more able to seek help and disclose information thus improving safety. |

with characteristics such as multiparity, age and ethnicity and presents an opportunity for those designing models of care to focus on these demographics. Where poor engagement with services is often associated with women's priorities and behaviours,[7 52 62] the women in this study highlighted system barriers. Interestingly, many women in this study felt that the ability to contact a known midwife anytime reduced the number of face-to-face appointments they needed. Where engagement in this study was measured through the number of antenatal appointments attended, it would be useful for future research to measure other forms of contact between women and healthcare professionals. Women from the hospital-based specialist model discussed the detrimental impact of midwives needing to cancel their appointments. Rather than a place-based issue this seems to relate to how the teams are organised as those in the hospital-based model were allocated one midwife who they saw for most of their appointments, whereas women accessing the community-based model described being cared for by the whole team. Finally, women from both community and hospital-based models expressed a preference for care to be based in the community or home setting as they felt it was more accessible and supportive of their needs.

The aim of both the group practice and specialist models evaluated appear to be being met with women more likely to receive more antenatal appointments with a known healthcare professional from these models. Women who received hospital-based antenatal care had significantly lower levels of continuity in the antenatal period. Women receiving care in the specialist model were more likely to be looked after in labour by a known healthcare professional compared with the other models. The qualitative data revealed this continuity of care lessened anxiety, the need to repeat often complex social and medical histories to numerous professionals and increased disclosure

of social risk factors, and preparedness for labour and birth, particularly for those who had the opportunity to meet the rest of the team. Multiparous, younger and black African women were more likely to miss antenatal appointments, as well as those with social risk factors. Black African women were also less likely to see a known healthcare professional than women of other ethnicities, perhaps reflecting an underlying mechanism of the health inequalities they experience.[63] There is a significant paucity of literature around black women's experiences of UK maternity care, but a US review[64] concluded that midwifery care can accommodate their specific needs through attentive provider, continuity of care and empowerment. The review stated that 'researchers must meet, consult with and listen to black women and hear their stories to understand the significance of midwifery to them. Only then, in partnership with, and based on suggestions for change from black women themselves, can health care providers and researchers begin to make changes in the health care system to facilitate improved antenatal care'.[64] This recommendation is as relevant to the UK context given the known stark inequalities.

### Strengths and limitations

Women interviewed during the study may have perceived the study questions to be testing them about their willingness to engage with their care. This limitation may have been lessened through the trust built between the participant and the researcher over the course of the longitudinal interviews.[65 66] The small and varied numbers in each quantitative data group should be taken into consideration due to the significant amount of multiple testing required to establish the separate effects of the model of care, place of care and service attended. This presents a potential limitation as the use of multiple testing can result in erroneous inferences, reducing the probability of detecting effects when they do exist.[67] This could be overcome in future research using larger sample sizes, and Bayesian analysis to test the apparent mitigating effects of the specialist models of care on inequalities in access and engagement.[68] That said, the claim to the causality of the mechanisms is strengthened as both qualitative and quantitative data analysis point to a causal link between the mechanism(s) and the outcomes studied.[68]

The analysis of quantitative data is also limited by the availability and depth of information routinely recorded in maternity services. It is recommended that more granular ethnicity categories, using the Office for National Statistics 18+ guidance[69] is used, as well as migration details such as country of birth, national identify, length of time in the UK, visa conditions and language proficiency. Finally, the generalisability of the findings is limited by the urban location of both specialist models of care evaluated and the UK's health system context. This is particularly significant when reflecting on the outcomes relating to place-based care what may have yielded significant outcomes in a densely populated, multicultural community, may yield very different results elsewhere. Research is needed to test the generalisability of the findings to rural and other community settings.

### CONCLUSION

This research highlights how carefully considered place-based care with a focus on continuity of carer can improve access and engagement with maternity services for women with social risk factors, but it is not a panacea. Hospital-based models of care with strict inclusion criteria may risk excluding women at increased risk who are yet to disclose social risk factors. Rather than the often-assumed 'relaxed maternal care-seeking behaviour' and 'women with social risk factors deprioritising maternity care' explanations for late booking and missed appointments, the findings highlight system barriers. Women want to be able to access care earlier in pregnancy, and there appears to be a lack of information regarding their choices in doing so, particularly for those who do not speak English or are unfamiliar with the system. The identification of specific mechanisms that improve access and engagement with services are outlined in the refined programme theory, which will enable those developing maternity services to structure models of care around women's individual needs.

**Acknowledgements** The authors would like to thank student midwives Laura Peazold, Mary Newman, Natalie Goodyear and Micaela Anthony for help with data collection and anonymisation, Justin Jagosh for invaluable advice on realist methodology and Sergio Silverio for support with qualitative data analysis.

**Contributors** HR-J, JMH, AH and JS contributed to the conceptualisation of the research question and methodology. EP and TG organised collection, anonymisation and analysis of the quantitative data. HR-J and KD analysed the quantitative data. HR-J collected qualitative data and analysed it with JMH. All authors interpreted the data analysis, read and approved the final manuscript. HR-J is the acting guarantor.

**Funding** This report is an independent research supported by the National Institute for Health Research (NIHR Doctoral Research Fellowship, HR-J, award no DRF-2017-10-033). AH is supported by the NIHR Applied Research Collaboration North Thames. JS (King's College London) is supported by the NIHR Applied Research Collaboration South London (NIHR ARC South London). JS is also an NIHR Senior Investigator. JMH is supported by a Post-doctoral Fellowship from Wellbeing of Women (Award Ref PRF006). KD is funded by the Medical Research Council (MRC) (grant number: MR/V005839/1). The views expressed in this article are those of the author(s) and not necessarily those of the National Health Service, the NIHR, the MRC or the Department of Health and Social Care.

**Competing interests** None declared.

**Patient and public involvement** Patients and/or the public were involved in the design, or conduct, or reporting, or dissemination plans of this research. Refer to the Methods section for further details.

**Patient consent for publication** Not applicable.

**Ethics approval** This research was approved by the London Brent Research Ethics Committee (HRA) REC Reference 18-LO-0701. Participants gave informed consent to participate in the study before taking part.

**Provenance and peer review** Not commissioned; externally peer reviewed.

**Data availability statement** Data are available upon reasonable request. Full quantitative data analysis attached as additional file or contact the lead author HR-J for qualitative and raw data files that are available upon reasonable request.

**ORCID iDs**
Hannah Rayment-Jones http://orcid.org/0000-0002-3027-8025
Kathryn Dalrymple http://orcid.org/0000-0003-0958-6725

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
