## [Reviewer comments · BMJ Open]

ARTICLE DETAILS

TITLE (PROVISIONAL)	Project20: Maternity care mechanisms that improve access and engagement for women with social risk factors in the UK. A mixed-methods, realist evaluation
AUTHORS	Rayment-Jones, Hannah; Dalrymple, Kathryn; Harris, James; Harden, Angela; Parslow, Elidh; Georgi, Thomas; Sandall, Jane

VERSION 1 – REVIEW

REVIEWER	Stephanie Brown Murdoch Childrens Research Institute, The University of Melbourne, Healthy Mothers Healthy Families, General Practice and Primary Care Academic Centre and Melbourne School of Population and Global Health
REVIEW RETURNED	18-Jun-2022

GENERAL COMMENTS	Thank you for inviting me to review this interesting and ambitious paper exploring the maternity care experiences of women with complex social risk factors. The authors are to be commended for the clarity of research questions, detailed explanation of quantitative findings, and the use of mixed methods to explore system level factors and mechanisms relating to access and engagement with care. Notwithstanding these strengths, there are some aspects of the paper that warrant further consideration to contextualise inferences. 1. The sample used for quant analyses comprised 799 women accessing different models of care at 2 urban inner city hospitals. Sample size calculations appear to be based on equal group sizes of 250 women, yet the numbers accessing each model of care varied significantly. Were any calculations undertaken to assess study power for analyses involving different group sizes and/or account for clustering in the data (given that analyses include use of multinomial regression accounting for clusters?) And what implications do these factors have for inferences in relation to data presented? 2. Social characteristics of samples: Table 1 and Table 2 use different ways to present information regarding ethnicity and migration status. I may have missed it, but is there an explanation for the different approaches used? It would be useful to have the same information for both samples, and also for women of refugee/migrant background to know length of time since arrival in UK, and English language proficiency/requirement for interpreter for the women included in the quantitative data analysis. If these data are not available in routinely collected data, I would encourage the authors to comment on this in the discussion, as it is clearly an
---

	important consideration for service level planning with regard to meeting the needs of these communities. 3. Adjustment for social and medical risk factors in Model. 1: If I understand correctly, adjustment was made for ethnicity, age, parity, deprivation score, number of social risk factors, and medical risk status. Given the relatively small numbers in some comparison groups, is there a risk of overadjustment in taking this approach? 4. Medical risk status: Apologies if I missed this, but how was medical risk defined at booking, and at labour commencement? And for analyses of model of care accessed by deprivation score and risk factors (Table 3) and gestation at booking in relation to model of care (Table 4), why adjust for medical risk status at labour commencement (assuming many factors influencing this are not apparent at the time of booking)? 5. Domestic abuse and mental health: How were these factors ascertained by services providing care to women in the quantitative sample? Arguably, the numbers reflect under ascertainment of these common experiences, including within specialist service settings? It would be useful for the authors to comment on this in drawing inferences from the findings. 6. Relational quality and missed appointments: These are complex analyses, and especially in Table 10, the small numbers in cells, present challenges with regard to inference. In Table 10, leaving aside the small number of women who missed 4 or more appointments, the direction of effects consistently suggest that women in standard care were more likely to miss appointments compared with women in specialist models. While caution is needed due to the small numbers, a more Bayesian approach to inference may be warranted here. 7. Longitudinal interview study – I may have missed this information, but I could not find reference to aspects of the design that reflect a longitudinal approach, or how this was managed in analysis. 8. Programme theory – Given the cultural diversity of the interview sample, I was surprised that no reference was made to cultural safety or access to interpreters as mechanisms to improve access and engagement. It is unclear whether these issues were considered in scope, or outside the framework of the refined programme theory? 9. Another aspect of the programme theory that I found curious is the assumption in outcome 4 that continuity of care will lead to greater identification of social risk in women who have not previously disclosed issues of concern. One element of the quantitative findings that might suggest otherwise is the very low level of ascertainment of domestic violence (<5% prevalence) and mental health issues (6% overall) in the routinely collected data. It would be interesting to examine these data in light of the programme theory to explore whether there was higher ascertainment in specialist models, and at what stage of pregnancy issues were identified (ie is there evidence that relational continuity results in greater recognition/higher disclosure of social risk factors at later gestations of pregnancy?). 10. Public and patient involvement – How were service users recruited and supported to engage in planning and development of
--	---

	the research? What was the reason for limiting their role in interpretation of data to the qualitative findings? 11. Strengths and limitations: I would encourage the authors to provide a more detailed account of strengths and limitations. What do they see as the main strengths of the approach they have taken? With regard to limitations, other factors to consider include: limitations of variables available in routine data, lack of generalisability outside UK health care system, sample size considerations outlined above. Minor issues: Table 1 (and several other tables) report p values of 0.000. Since probability cannot be zero, suggest report as $p < 0.001$ Page 5, line 48. Typo in the sentence “One model based is”, should this be “one model is based”? Page 16, line 30 (and in other places) Data are plural so verb is ‘were’ Table 12 – heading says “qualitative” outcomes. Should this be ‘quantitative’?
--	---

REVIEWER	Ferdinand Mukumbang University of the Western Cape, School of Public Health
REVIEW RETURNED	16-Oct-2022

GENERAL COMMENTS	Dear Editor, I want to thank the authors for such a thoughtful and well-put-together mixed methods realist study. I was particularly impressed by how much detail the authors put into obtaining the quantitative and qualitative pieces of information towards their realist/mechanism-based theory formulation. Despite the above strengths of the paper. There is a major weakness. While the authors explicitly illustrated how information informing the theory formulation was contained, the process by which the theory was formulated is not explained. From the information from the qualitative and quantitative sources, I have no idea how the authors obtained their CMO configurations illustrated in Table 13. I think that is a major flaw in the methodology of the paper. Formulating theories in realist research is known as retroductive theorizing, but there was no mention of this in this otherwise well-thought-out paper. The authors should consider these three resources to help them in this aspect: Mukumbang, F. C., Kabongo, E. M., & Eastwood, J. G. (2021). Examining the Application of Retroductive Theorizing in Realist-Informed Studies. International Journal of Qualitative Methods, 20. https://doi.org/10.1177/16094069211053516 Mukumbang, F. C. (2021). Retroductive Theorizing: A Contribution of Critical Realism to Mixed Methods Research. Journal of Mixed Methods Research. https://doi.org/10.1177_15586898211049847
---

	Justin Jagosh (2020) Retroductive theorizing in Pawson and Tilley's applied scientific realism, Journal of Critical Realism, 19:2, 121-130, DOI: 10.1080/14767430.2020.1723301 The authors should also consider using a causal-loop diagram to illustrate their theories. In that way, the reader can appreciate the complexity of how women access and engagement with maternity care. See an example of how this was done: Mukumbang FC, De Souza D, Liu H, et al. Unpacking the design, implementation and uptake of community-integrated health care services: a critical realist synthesis BMJ Global Health 2022;7:e009129.
--	---

VERSION 1 – AUTHOR RESPONSE

Reviewer 1 Dr. Stephanie Brown, Murdoch Childrens Research Institute, The University of Melbourne, South Australian Health and Medical Research Council	
Thank you for inviting me to review this interesting and ambitious paper exploring the maternity care experiences of women with complex social risk factors. The authors are to be commended for the clarity of research questions, detailed explanation of quantitative findings, and the use of mixed methods to explore system level factors and mechanisms relating to access and engagement with care. Notwithstanding these strengths, there are some aspects of the paper that warrant further consideration to contextualise inferences.	Thank you for your kind and encouraging feedback. Your detailed insights have been incredibly helpful in strengthening this manuscript. Please see revisions made below that we hope will contextualise the findings and inferences, and increase transparency of the limitations of this research.
1. The sample used for quant analyses comprised 799 women accessing different models of care at 2 urban inner city hospitals. Sample size calculations appear to be based on equal group sizes of 250 women, yet the numbers accessing each model of care varied significantly. Were any calculations undertaken to assess study power for analyses involving different group sizes and/or account for clustering in the data (given that analyses include use of	Thank you for this important consideration. As the power calculations were undertaken prior to data access and prospective analysis there was an assumption (based on national targets for women to have access to continuity of care models) that the groups would be more evenly distributed. In reality, as you point out, the number of women allocated to different models of care varied significantly, with many outcomes being underpowered. This has been highlighted as a limitation of this research and prioritised as a future research recommendation (pg 4): 'The relatively small and varied numbers in each quantitative data group, and the multiple testing required

multinomial regression accounting for clusters?) And what implications do these factors have for inferences in relation to data presented?	to establish the separate effects of the model of care, place of care and service attended may increase the risk of erroneous inferences.' We have also revised the discussion section, highlighting that the quantitative analysis should be reviewed with caution due to the potentially underpowered groups, and not without the insights of the qualitative data to infer mechanisms, in line with realist underpinnings ¹ (pg28): The small and varied numbers in each quantitative data group should be taken into consideration due to the significant amount of multiple testing required to establish the separate effects of the model of care, place of care and service attended. This presents a potential limitation as the use of multiple testing can result in erroneous inferences, reducing the probability of detecting effects when they do exist ².
2. Social characteristics of samples: Table 1 and Table 2 use different ways to present information regarding ethnicity and migration status. I may have missed it, but is there an explanation for the different approaches used? It would be useful to have the same information for both samples, and also for women of refugee/migrant background to know length of time since arrival in UK, and English language proficiency/requirement for interpreter for the women included in the quantitative data analysis. If these data are not available in routinely collected data, I would encourage the authors to comment on this in the discussion, as it is clearly an important consideration for service level planning with regard to meeting the needs of these communities.	Thank you. The reason for the differences in how these are presented are that we were restricted to the quantitative data that is recorded in the routinely collected hospital data across the two different services. For the qualitative analysis we were able to collect more detailed demographics. Language proficiency has been added to the table. Added to discussion section (pg29): 'The analysis of quantitative data is also limited by the availability and depth of information routinely recorded in maternity services. It is recommended that more granular ethnicity categories, using the ONS 18+ guidance ³ is used, as well as migration details such as country of birth, national identify, length of time in the UK, visa conditions and language proficiency.'
3. Adjustment for social and medical risk factors in Model. 1: If I understand correctly, adjustment was made for ethnicity, age, parity, deprivation score, number of social risk factors, and medical risk status. Given the relatively small numbers in some comparison groups, is there a risk of overadjustment in taking this approach?	Yes the authors agree that this is a risk and have commented on this limitation, including in the highlighted strengths and limitations section. Full adjusted and unadjusted data analysis is provided in supplementary file 2. Again, it is highlighted in the revised manuscript (pg 29): The small and varied numbers in each quantitative data group should be taken into consideration due to the significant amount of multiple testing required to establish the separate effects of the model of care, place of care and service attended. This presents a potential limitation as the use of multiple testing can result in erroneous

	inferences, reducing the probability of detecting effects when they do exist². This could be overcome in future research using larger sample sizes, and Bayesian analysis to test the apparent mitigating effects of the specialist models of care on inequalities in access and engagement¹. That said, the claim to the causality of the mechanisms is strengthened as both qualitative and quantitative data analysis point to a causal link between the mechanism(s) and the outcomes studied¹.
4. Medical risk status: Apologies if I missed this, but how was medical risk defined at booking, and at labour commencement? And for analyses of model of care accessed by deprivation score and risk factors (Table 3) and gestation at booking in relation to model of care (Table 4), why adjust for medical risk status at labour commencement (assuming many factors influencing this are not apparent at the time of booking)?	Revised- 'Medical risk status' added to Table 1. When referring to medical risk status we have now directed the reader to supplementary file 1 'definitions' where medical risk status is recorded as high or low by the healthcare professional at the initial maternity booking appointment, and then again at the onset of labour (pg 9). The decision to adjust for medical risk factors at both booking and onset of labour is to account for those pregnancies which began as low risk but over the course of the pregnancy became high risk that would change the appropriate number of appointments and level of engagement with services.
5. Domestic abuse and mental health: How were these factors ascertained by services providing care to women in the quantitative sample? Arguably, the numbers reflect under ascertainment of these common experiences, including within specialist service settings? It would be useful for the authors to comment on this in drawing inferences from the findings.	Revised definitions in supplementary file 1: 'these risk factors could be ascertained through self-reporting or previously recorded data'. Added to discussion: (pg 26) 'Previous research has found a correlation between continuity models of care and increased referrals to support services⁴. but it must be acknowledged that under ascertainment of sensitive issues such as mental health and domestic abuse remains likely whilst women perceive services as a form of surveillance and risk⁵.' and (pg28): 'The qualitative data revealed this continuity of care lessened anxiety, the need to repeat often complex social and medical histories to numerous professionals, and increased disclosure of social risk factors...'
6. Relational quality and missed appointments: These are complex analyses, and especially in Table 10, the small numbers in cells, present challenges with regard to inference. In	Thank you for this point and suggestion of Bayesian analysis. This has prompted further reading has been put forward as a future research recommendation within the discussion (pg28): 'This could be overcome in future research using larger sample sizes, and Bayesian

Table 10, leaving aside the small number of women who missed 4 or more appointments, the direction of effects consistently suggest that women in standard care were more likely to miss appointments compared with women in specialist models. While caution is needed due to the small numbers, a more Bayesian approach to inference may be warranted here.	analysis to test the apparent mitigating effects of the specialist models of care on inequalities in access and engagement¹.’ in line with similar advice such as ‘The dominance of qualitative approaches in the field of realistic evaluation has been the starting point of several recent methodological papers promoting a more quantitative turn in realistic evaluation and in theory-based evaluation (e.g. Ford et al., 2018; Giffoni et al., 2018; Hawkins, 2016). The methods promoted include ‘The Bayesian Network Approach’ (Giffoni et al., 2018), ‘Process Tracing and Bayesian updating’ (Befani and Stedman-Bryce, 2017) and ‘Structural Equation Modelling’ (Ford et al., 2018). These approaches are, however, quite advanced and not readily applicable for evaluators with only rudimentary statistical training.’¹
7. Longitudinal interview study – I may have missed this information, but I could not find reference to aspects of the design that reflect a longitudinal approach, or how this was managed in analysis.	Revised. Added to methods (pg 8): ‘Semi-structured, longitudinal interviews with 20 women with low socioeconomic status and social risk factors who were receiving specialist care from one of the two service providers were carried out at approximately 28 and 36 weeks’ gestation, and 6 weeks post birth’, and (pg 9): ‘The qualitative data were coded using NVivo v.12 and analysed using a thematic framework analysis⁶. This allowed for the organisation of a large qualitative dataset into a coding framework developed using previously constructed programme theories^{7,8}, to uncover new theories and differences in women’s experiences depending on their characteristics⁶.’ and (pg 9):‘This method suited the longitudinal approach to data collection as changes in women’s perceptions and relationships with healthcare providers could be seen over the course of their pregnancy and postnatal period.’
8. Programme theory – Given the cultural diversity of the interview sample, I was surprised that no reference was made to cultural safety or access to interpreters as mechanisms to improve access and engagement. It is unclear whether these issues were considered in scope, or outside the framework of the refined programme theory?	Yes this is an excellent point. As the issue of language barriers and use of interpreter services was such a significant issue for these women it was tested in a separate analysis of additional programme theories generated by a previous realist synthesis⁹. The refined programme theory is published here: https://equityhealthj.biomedcentral.com/articles/10.1186/s12939-021-01570-8 We have revised the manuscript to clarify this in the discussion (pg 26): ‘The wider Project20 evaluation previously tested programme theory relating to interpreter services for pregnant women with social risk factors, finding that despite accessing a specialist model of care

	women experienced a lack of regulation and access to high-quality interpretation services ¹⁰.’
9. Another aspect of the programme theory that I found curious is the assumption in outcome 4 that continuity of care will lead to greater identification of social risk in women who have not previously disclosed issues of concern. One element of the quantitative findings that might suggest otherwise is the very low level of ascertainment of domestic violence (<5% prevalence) and mental health issues (6% overall) in the routinely collected data. It would be interesting to examine these data in light of the programme theory to explore whether there was higher ascertainment in specialist models, and at what stage of pregnancy issues were identified (ie is there evidence that relational continuity results in greater recognition/higher disclosure of social risk factors at later gestations of pregnancy?).	Apologies, this needs clarifying. The findings summarised in the refined programme theory (table 13) refer to where care is situated and who it reaches rather than increased disclosure. So if services are situated in areas of deprivation, according to the quantitative findings that increased deprivation score correlates to increased social risk factors, services situated in deprived areas will be more likely to provide an enhanced level of care to women with this increased risk whether the women have disclosed their issues or not. This supports an ⁴ earlier research finding that women under a continuity model of care go on to receive more referrals to support services. The wider evaluation tests the theory that women then go on to feel more able to disclose sensitive issues, and receive an enhanced level of specialist support. Since this review process this paper has now been published in Women and Birth ⁵. This has now been clarified in this manuscript’s discussion (pg 26): Previous research has found a correlation between continuity models of care and increased disclosure and referral to support services ^{4 5}, but it must be acknowledged that under ascertainment of sensitive issues such as mental health and domestic abuse remains likely whilst women perceive services as a form of surveillance and risk ⁵.
10. Public and patient involvement – How were service users recruited and supported to engage in planning and development of the research? What was the reason for limiting their role in interpretation of data to the qualitative findings?	Revised (pg 7): Multiple representative, diverse groups of service users were involved in the planning and development of this research. They were recruited through local community groups, clinicians, and existing patient involvement groups. Using participatory appraisal methods and online engagement events, recent maternity service users provided feedback on the protocol, study materials, interview guides, and refinement of programme theories. They also prioritised outcome measures and reviewed the qualitative data analysis. Training needs were identified by the service users for analysis of quantitative data and further research addressing maternal health inequities ¹¹.
11. Strengths and limitations: I would	

encourage the authors to provide a more detailed account of strengths and limitations. What do they see as the main strengths of the approach they have taken? With regard to limitations, other factors to consider include: limitations of variables available in routine data, lack of generalisability outside UK health care system, sample size considerations outlined above.	Revised extensively in both the discussion (pg 28) and summary of strengths and limitations of this research (pg 4 after abstract) with particular reference to the sample size and future Bayesian analysis recommendations:  • The specialist models of care evaluated in this study are situated in the UK’s complex maternity system and can be appropriately investigated using a mixed-methods, realist evaluation design. • Both qualitative and quantitative methods are used to understand not only if access and engagement are improved by specialist models of care, but how and in what context. • Longitudinal interviews were undertaken to increase trust between participants and the researcher and lessen the perception of surveillance. • ‘The relatively small and varied numbers in each quantitative data group, and the multiple testing required to establish the separate effects of the model of care, place of care and service attended might result in a change in statistical power, reducing the probability of detecting effects when they do exist.’ • The generalisability of the findings is limited by the urban location of both specialist models of care evaluated and the UK health system context.
Minor issues: Table 1 (and several other tables) report p values of 0.000. Since probability cannot be zero, suggest report as $p < 0.001$ Page 5, line 48. Typo in the sentence “One model based is”, should this be “one model is based”? Page 16, line 30 (and in other places) Data are plural so verb is ‘were’ Table 12 – heading says “qualitative” outcomes. Should this be ‘quantitative’?	Revised throughout manuscript Revised Revised throughout manuscript

	Revised
Reviewer 2 Dr. Ferdinand Mukumbang, University of the Western Cape, Ferdinand C. Mukumbang	
I want to thank the authors for such a thoughtful and well-put-together mixed methods realist study. I was particularly impressed by how much detail the authors put into obtaining the quantitative and qualitative pieces of information towards theory their realist/mechanism-based theory formulation.	Thank you for your time reviewing this manuscript and very helpful and encouraging feedback. Please see revisions made below.
Despite the above strengths of the paper. There is a major weakness. While the authors explicitly illustrated how information informing the theory formulation was contained, the process by which the theory was formulated is not explained. From the information from the qualitative and quantitative sources, I have no idea how the authors obtained their CMO configurations illustrated in Table 13. I think that is a major flaw in the methodology of the paper. Formulating theories in realist research is known as retroductive theorizing, but there was no mention of this in this otherwise well-thought-out paper. The authors should consider these three resources to help them in this aspect: Mukumbang, F. C., Kabongo, E. M., & Eastwood, J. G. (2021). Examining the Application of Retroductive Theorizing in Realist-Informed Studies. International Journal of Qualitative Methods, 20. https://doi.org/10.1177/16094069211053516	Thank you for noting this and for the very useful resources that have been referenced appropriately in the manuscript. The initial programme theories were developed using retroductive theorising in a previous realist synthesis⁹ and focus groups with midwives¹² before being tested in this aspect of the research. We have revised supplementary file 2 to clearly show the process of theory refinement between the initial programme theories and the CMO configuration described in table 13. We have also revised the methods section and included figure 1 to visually demonstrate the process of refinement that is then detailed in supplementary file 2. See pg 6: The aims of this study were approached through the testing and refinement of initial programme theories constructed in an earlier synthesis of literature⁷ and focus groups with midwives⁸ relating to how women with social risk factors access and engage with maternity care- see figure 1 for the theory refinement process. Retroductive theorizing was used to uncover meaningful causal mechanisms, often focusing on how the wider context and human response to different aspects of maternity care leads to specific outcomes¹³. This approach offers an epistemologically and ontologically grounded way of integrating mixed methods, often analysing qualitative data to find the causal relationship behind quantitative

Mukumbang, F. C. (2021). Retroductive Theorizing: A Contribution of Critical Realism to Mixed Methods Research. Journal of Mixed Methods Research. https://doi.org/10.1177_15586898211049847 Justin Jagosh (2020) Retroductive theorizing in Pawson and Tilley's applied scientific realism, Journal of Critical Realism, 19:2, 121-130, DOI: 10.1080/14767430.2020.1723301	findings ¹⁴.
The authors should also consider using a causal-loop diagram to illustrate their theories. In that way, the reader can appreciate the complexity of how women access and engagement with maternity care. See an example of how this was done: Mukumbang FC, De Souza D, Liu H, et al. Unpacking the design, implementation and uptake of community-integrated health care services: a critical realist synthesis. BMJ Global Health 2022;7:e009129.	Thank you for this helpful suggestion and clear example. Based on this the authors are in the process of creating a causal loop diagram for the wider Project20 evaluation findings to demonstrate the complexity of maternity care for this population as a whole. This will complement an infographic that has been produced for policy relevance. The causal loop diagram will be presented at the ICM conference 2023.